# A Spatially-Aware Multiple Instance Learning Framework for Digital Pathology

**Hassan Keshvarikhojasteh**[1]                    H.KESHVARIKHOJASTEH@TUE.NL
**Mihail Tifrea**[2]                                M.TIFREA@STUDENT.TUE.NL
**Sibylle Hess**[2]                                       S.C.HESS@TUE.NL
**Josien P.W. Pluim**[1]                                   J.PLUIM@TUE.NL
**Mitko Veta**[1]                                          M.VETA@TUE.NL

[1] *Department of Biomedical Engineering, Eindhoven University of Technology, Eindhoven, The Netherlands*

[2] *Mathematics and Computer Science Department, Eindhoven University of Technology, Eindhoven, The Netherlands*

**Editor:**

## Abstract

Multiple instance learning (MIL) is a promising approach for weakly supervised classification in pathology using whole slide images (WSIs). However, conventional MIL methods such as Attention-Based Deep Multiple Instance Learning (ABMIL) typically disregard spatial interactions among patches that are crucial to pathological diagnosis. Recent advancements, such as Transformer based MIL (TransMIL), have incorporated spatial context and inter-patch relationships. However, it remains unclear whether explicitly modeling patch relationships yields similar performance gains in ABMIL, which relies solely on Multi-Layer Perceptrons (MLPs). In contrast, TransMIL employs Transformer-based layers, introducing a fundamental architectural shift at the cost of substantially increased computational complexity. In this work, we enhance the ABMIL framework by integrating interaction-aware representations to address this question. Our proposed model, Global ABMIL (GABMIL), explicitly captures inter-instance dependencies while preserving computational efficiency. Experimental results on two publicly available datasets for tumor subtyping in breast and lung cancers demonstrate that GABMIL achieves up to a 7 percentage point improvement in AUPRC and a 5 percentage point increase in the Kappa score over ABMIL, with minimal or no additional computational overhead. These findings underscore the importance of incorporating patch interactions within MIL frameworks. Our code is available at GABMIL.

**Keywords:**   Multiple instance learning (MIL), whole slide images (WSIs), Attention-Based Deep MIL (ABMIL), Transformer based MIL (TransMIL), GlobaL Attention-Based Deep MIL (GABMIL).

## 1 Introduction

Digital pathology involves performing pathology practices within a fully digital workflow. In this setting, pathologists analyze high-resolution Whole Slide Images (WSIs) on digital screens rather than traditional microscopes (Aeffner et al., 2019). WSIs are high-resolution digital scans of tissue sections, acquired using digital slide scanners. Transition to digital pathology offers several advantages, including improved workflow efficiency, enhanced

patient safety, higher diagnostic consistency, and the potential to mitigate workforce shortages (Williams et al., 2017). Moreover, digital pathology opens up opportunities for advanced image analysis techniques, facilitating computational pathology.

However, the extremely large size of WSIs introduces a significant computational bottleneck, making it infeasible to process them directly with deep neural networks. Therefore, they are typically divided into smaller patches that can be individually processed. Another challenge arises from the labeling process, as annotations are usually provided at the slide level rather than for individual patches, leading to a weakly supervised learning scenario (Kanavati et al., 2020). Consequently, weakly supervised learning methods have emerged as an effective strategy for handling WSI classification (Zhou, 2018).

Multiple instance learning (MIL) serves as a powerful tool for addressing weakly supervised classification in digital pathology. In MIL-based approaches, a WSI is represented as a collection of patches (bag), where each patch is encoded as a feature embedding. These embeddings are then aggregated into a single slide-level representation for classification. However, current MIL methods often overlook interactions between distinct instances. Despite some enhancements in various tasks, this limitation is not universally applicable. In reality, pathologists typically consider both local and global contextual information when making diagnostic decisions. Consequently, it is intriguing to asses the impact of inter-instance dependencies for MIL networks.

Attention-Based Deep Multiple Instance Learning (ABMIL) (Ilse et al., 2018) is a widely used MIL method that aggregates instances embeddings using a weighted sum, where the weights are assigned using a novel attention mechanism. This approach highlights diagnostically relevant instances while diminishing the contribution of less informative ones. The simplicity of ABMIL originates from the usage of only Multi-Layer Perceptrons (MLPs) in the model's design. However, a major drawback of ABMIL is its lack of using spatial information within the image.

Transformer based MIL (TransMIL) (Shao et al., 2021) addresses this limitation by integrating spatial information among patches. Specifically, TransMIL develops a new positional encoding scheme called Pyramidal Position Encoding Generator (PPEG), paired with the Transformer architecture (Vaswani et al., 2017), achieving superior performance compared to previous MIL approaches. Specifically, TransMIL consists of two multi-head self-attention layers and a PPEG layer in between. However, the main drawback of the TransMIL is its high computational complexity associated with running self-attention over the entire embedding space. Furthermore, the impact of patch interactions in ABMIL remains unexplored, as TransMIL introduces a fundamental architectural shift compared to ABMIL.

To address this gap, we suggest Global ABMIL (GABMIL), an enhanced ABMIL framework that incorporates inter-instance interaction while maintaining computational efficiency. Our approach builds upon recent advancements in Vision Transformers (ViTs) (Tu et al., 2022; Tolstikhin et al., 2021), specifically integrating BLOCK and GRID attention modules (Tu et al., 2022) to capture global relationships among instances. Unlike Transformer-based approaches, we develop a MLP-based architecture inspired by (Tolstikhin et al., 2021), allowing us to incorporate spatial context while preserving computational efficiency. We evaluate our method on two publicly available datasets for tumor subtyping, demonstrating that GABMIL significantly improves classification performance

Figure 1: An overview of our GABMIL method. We first divide the input WSI into patches and extract their corresponding features using a pretrained model. The Spatial Information Mixing Module (SIMM) then integrates spatial information into the feature representations. Finally, the ABMIL model predicts the slide-level label.

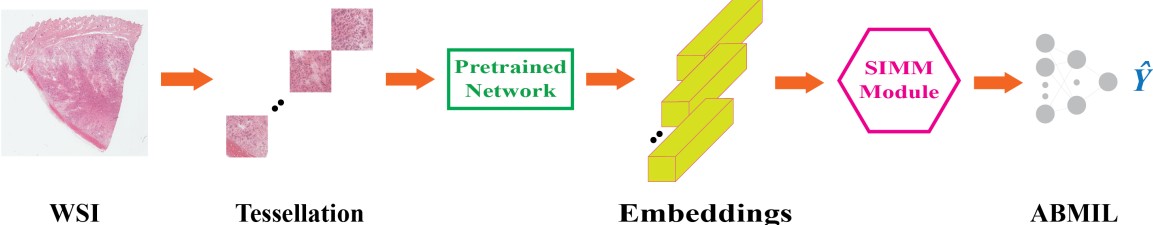

compared to ABMIL, with no or minimal computational increase, while TransMIL incurs substantially higher computational costs.

## 2 Method

### 2.1 Problem Definition

For our analysis, we follow established methodologies by first tessellating the input slide into $N$ small patches and then extracting features from these patches using a pretrained model. In the final step, the slide label ($\hat{Y}$) is predicted using an aggregation module.

### 2.2 Method Overview

**Global ABMIL (GABMIL)** involves incorporating spatial information into the embeddings using Spatial Information Mixing Module (SIMM). Once the spatial information is integrated, ABMIL network is used to predict the slide label. An overview of the entire architecture is provided in Fig. 1.

**Spatial Information Mixing Module (SIMM)** begins by reorganizing the embeddings into their original two-dimensional spatial layout, corresponding to the whole-slide image from which they were extracted. Specifically, we reshape our tensor of shape $(N, C)$ (where $N$ is the number of embeddings and $C$ is the embedding dimension) into a tensor of shape $(W, H, C)$, with $W$ and $H$ representing the width and height of the spatial grid, respectively. This restructuring enables the incorporation of spatial information by leveraging the positional details inherent in the original slide layout.

To integrate this information into the embeddings, we draw inspiration from the BLOCK and GRID attention modules of the MaxViT architecture (Tu et al., 2022). To apply the BLOCK attention, we first transform our $(W, H, C)$ tensor into a $(\frac{\hat{H}}{P} \times \frac{\hat{W}}{P}, P \times P, C)$ shaped tensor where $\hat{W}$ and $\hat{H}$ represent the augmented width and height, padded with zeros to ensure that $W$ and $H$ are divisible by the window size $P$. Then, we use a MLP shared across all windows instead of the self-attention mechanism to alleviate the computational overhead of self-attention and simplify the architecture. This MLP is applied solely on the second dimension of the reshaped $(\frac{\hat{H}}{P} \times \frac{\hat{W}}{P}, P \times P, C)$ tensor, effectively acting as a depth-

Figure 2: (a) Illustration of the SIMM (BOTH configuration). Patch features are reposi­tioned according to their original spatial arrangement. The BLOCK and GRID attention modules are then applied sequentially to integrate spatial information into the feature representations. (b) The BLOCK attention module captures spa­tial information within partitioned windows using a MLP layer. (c) The GRID attention module models spatial information within each partitioned grid using a MLP layer.

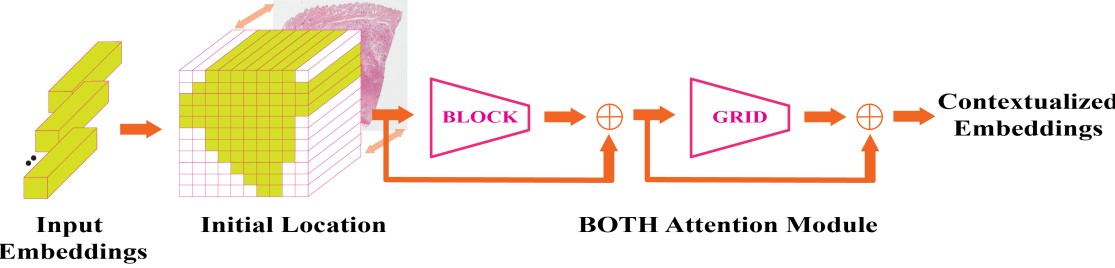

**(a) Spatial Information Mixing Module**

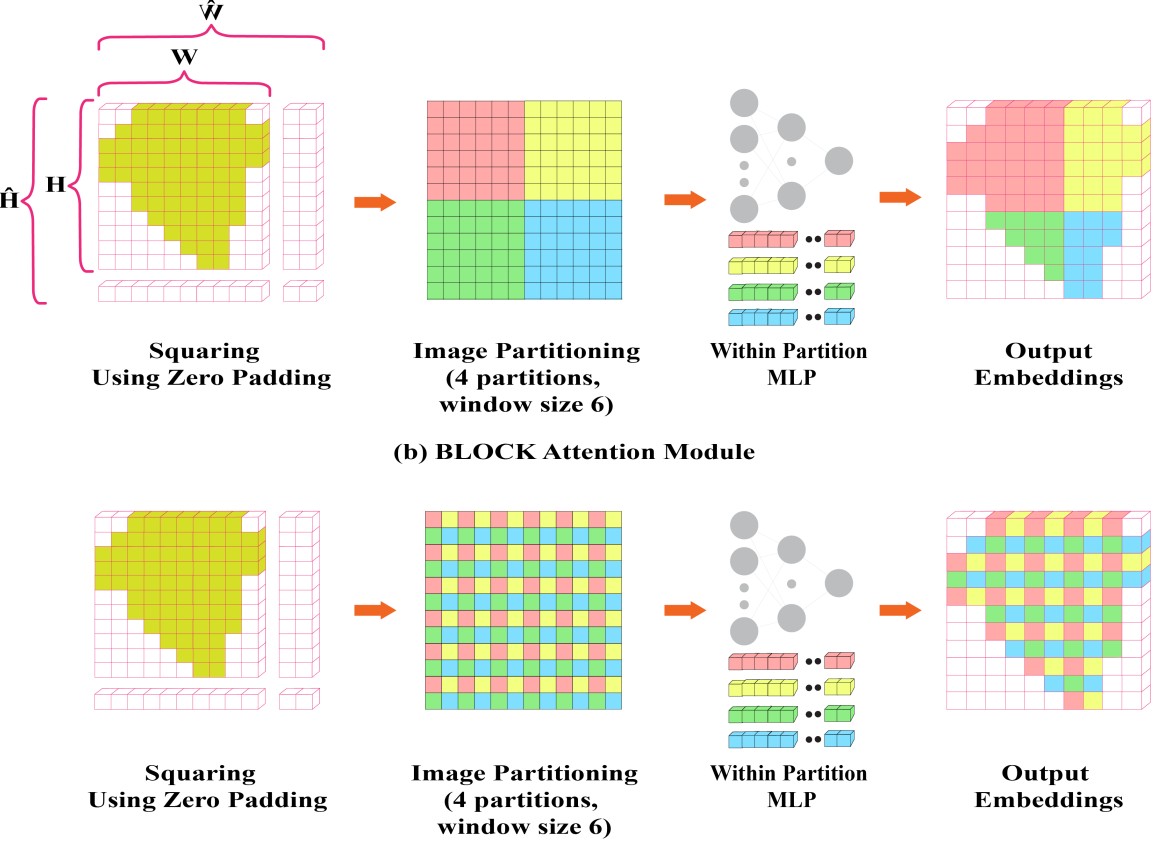

**(b) BLOCK Attention Module**

**(c) GRID Attention Module**

Table 1: Slide-Level Classification Evaluation on TCGA BRCA dataset using ImageNet pretrained ResNet50 to extract instance features. The values are reported as mean ± standard deviation. The best ones are in bold. The flops are measured with 120 instances per bag, and the instance feature extraction is not considered in the presented flops.

| Model | AUC | F1 | Recall | Kappa | AUPRC | FLOPs |
|---|---|---|---|---|---|---|
| ABMIL | 0.88±0.05 | 0.78±0.06 | 0.78±0.07 | 0.57±0.12 | 0.67±0.11 | 94M |
| TRANSMIL | 0.89±0.05 | 0.77±0.06 | 0.77±0.08 | 0.55±0.12 | 0.71±0.11 | 614M |
| **BLOCK_3** | **0.91±0.04** | **0.81±0.05** | **0.80±0.07** | **0.62±0.1** | **0.74±0.09** | **94M** |
| GRID_4 | 0.90±0.04 | 0.79±0.04 | 0.78±0.05 | 0.59±0.07 | 0.72±0.10 | 94M |
| BOTH_4 | 0.89±0.05 | 0.79±0.08 | 0.78±0.08 | 0.58±0.16 | 0.71±0.15 | 94M |

Table 2: Slide-Level Classification Evaluation on TCGA BRCA dataset with sef-supervised pretrained ResNet50 to extract instance features. The values are reported as mean ± standard deviation. The best ones are in bold.

| Model | AUC | F1 | Recall | Kappa | AUPRC | FLOPs |
|---|---|---|---|---|---|---|
| **ABMIL** | 0.92±0.04 | 0.82±0.06 | 0.80±0.08 | 0.64±0.12 | 0.79±0.11 | **94M** |
| TRANSMIL | 0.91±0.05 | 0.79±0.08 | 0.79±0.09 | 0.58±0.16 | 0.72±0.15 | 614M |
| **BLOCK_5** | **0.93±0.04** | **0.85±0.05** | **0.83±0.06** | **0.69±0.09** | **0.79±0.12** | 95M |
| GRID_5 | 0.92±0.05 | 0.83±0.06 | 0.83±0.08 | 0.67±0.12 | 0.78±0.13 | 95M |
| BOTH_6 | 0.93±0.04 | 0.83±0.04 | 0.82±0.06 | 0.65±0.7 | 0.79±0.11 | 97M |

wise convolution where all the filters are replaced with a single MLP. This modification is inspired by the Mixer (Tolstikhin et al., 2021), which demonstrates the effectiveness of MLPs in sharing spatial information and achieves comparable results to state-of-the-art models on image classification benchmarks, all while using a simpler architecture consisting of only MLPs.

Similarly, To apply the GRID attention, we reshape the initial $(W, H, C)$ tensor into a $(G \times G, \frac{\hat{H}}{G} \times \frac{\hat{W}}{G}, C)$ shaped tensor, using a $G \times G$ uniform grid. Next, we apply a MLP on the first dimension.

We evaluate three design variants of our SIMM. First, we use only the BLOCK attention module. Second, we use only the GRID module. Finally, we integrate both approaches sequentially (denoted as BOTH), where the BLOCK module is followed by the GRID module, aligning with the ordering strategy adopted in MaxViT. To preserve the original morphological details encoded in each embedding, we use a residual connection for whole designs. A schematic representation of these architectures is displayed in Fig. 2. The performance of all three configurations is assessed in the results section.

Figure 3: Representation of a slide from TCGA BRCA and its corresponding average slide-level feature maps across channels using different attention modules. From left to right: the original slide, the slide-level feature map generated using ImageNet pretrained extracted patch features, the output slide-level feature map after applying the corresponding attention module, and the contextualized slide-level feature map after the residual connection is applied.

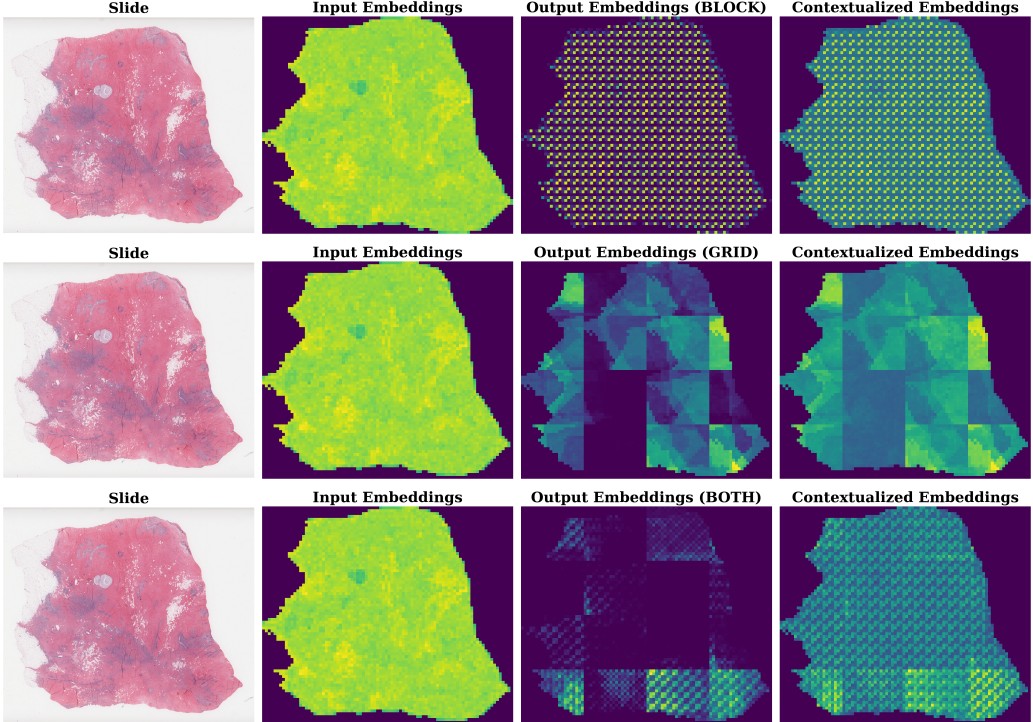

## 3 Experiments

### 3.1 Datasets

We use two publicly available WSI datasets. The **TCGA Breast Invasive Carcinoma Cancer (TCGA BRCA)** dataset includes 1038 H&E WSIs, representing two types of breast cancer; Invasive Ductal Carcinoma (IDC) versus Invasive Lobular Carcinoma (ILC) with slide-level labels. The **TCGA Lung Cancer (TCGA LUNG)** dataset is a public dataset for Non-Small Cell Lung Carcinoma (NSCLC) subtyping, containing 1046 H&E slides for Lung Adenocarcinoma (LUAD) and Lung Squamous Cell Carcinoma (LUSC), also with slide-level labels.

### 3.2 Implementation Details

For dataset preprocessing, we extract features from non-overlapping 256×256 patches at 10× magnification using the ImageNet and self-supervised [1] pretrained ResNet50. The

---

1. We used the weights provided by (Kang et al., 2023).

Figure 4: AUPRC scores of the three design variants of SIMM (BLOCK, GRID, and BOTH ) on TCGA BRCA dataset using ImageNet pretrained extracted features.

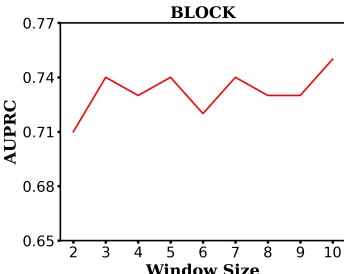 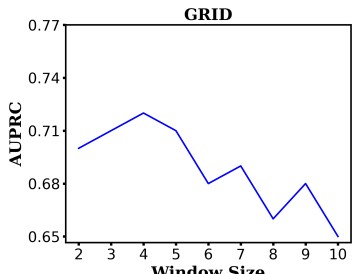 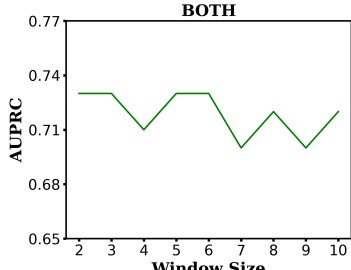

Table 3: Slide-Level Classification Evaluation on TCGA LUNG dataset using ImageNet pretrained ResNet50 to extract instance features. The values are reported as mean ± standard deviation. The best ones are in bold. The flops are measured with 120 instances per bag, and the instance feature extraction is not considered in the presented flops.

| Model | AUC | F1 | Recall | Kappa | AUPRC | FLOPs |
|---|---|---|---|---|---|---|
| ABMIL | 0.93±0.02 | 0.85±0.04 | 0.85±0.04 | 0.70±0.08 | 0.91±0.05 | 94M |
| TRANSMIL | 0.94±0.02 | 0.86±0.03 | 0.86±0.03 | 0.72±0.06 | 0.93±0.02 | 614M |
| **BLOCK_2** | 0.94±0.01 | 0.86±0.03 | 0.86±0.03 | 0.72±0.07 | **0.93±0.03** | 94M |
| GRID_2 | 0.94±0.02 | 0.85±0.05 | 0.85±0.04 | 0.71±0.09 | 0.93±0.03 | 94M |
| **BOTH_2** | **0.94±0.02** | **0.87±0.04** | **0.87±0.04** | **0.74±0.08** | 0.92±0.04 | **94M** |

extracted features dimensionality is 1024, and these features are subsequently compressed to a size of 512. The models are trained for 50 epochs using the Adam optimizer with a learning rate of 1e-4 and a weight decay of 1e-3, except for TransMIL, where a weight decay of 1e-2 is used to mitigate overfitting. Test results are reported using the area under the curve (AUC) and area under the precision-recall curve (AUPRC) metrics, calculated via 10-fold cross-validation. Additionally, the slide-level F1 score, Recall, and Kappa scores are considered, with a threshold of 0.5. The optimal window size is selected from the range 1 to 10 based on the validation loss. For convenience, we use abbreviated notation to indicate the window size and SIMM configuration; for example, BLOCK_3 refers to the BLOCK attention module with a window size of 3.

### 3.3 Results

Table 1 presents the WSI classification results on the TCGA BRCA dataset using ImageNet-pretrained extracted features. Our best model, BLOCK_3, achieved a 2 percentage point improvement in AUC, 3 percentage point in F1 score, 2 percentage point in Recall, 5 percentage point in Kappa, and 3 percentage point in AUPRC compared to the second-best method, while maintaining the computational efficiency of ABMIL. Similarly, when using self-supervised features (Table 2), our best model, BLOCK_5, demonstrated a 1 percentage

Table 4: Slide-Level Classification Evaluation on TCGA LUNG dataset with sef-supervised pretrained ResNet50 to extract instance features. The values are reported as mean ± standard deviation. The best ones are in bold.

| Model | AUC | F1 | Recall | Kappa | AUPRC | FLOPs |
|---|---|---|---|---|---|---|
| ABMIL | 0.96±0.02 | 0.88±0.03 | 0.88±0.03 | 0.76±0.06 | 0.95±0.02 | 94M |
| TRANSMIL | 0.95±0.02 | 0.87±0.03 | 0.87±0.03 | 0.74±0.07 | 0.95±0.02 | 614M |
| **BLOCK_2** | **0.96±0.02** | **0.89±0.03** | **0.89±0.03** | **0.78±0.06** | **0.96±0.02** | **94M** |
| GRID_1 | 0.96±0.01 | 0.88±0.03 | 0.88±0.03 | 0.76±0.06 | 0.96±0.02 | 94M |
| BOTH_4 | 0.95±0.02 | 0.87±0.03 | 0.87±0.02 | 0.73±0.07 | 0.94±0.03 | 95M |

point increase in AUC, 3 percentage point in F1 score, 3 percentage point in Recall, and 5 percentage point in Kappa, with only a marginal increase in computational cost compared to ABMIL. Furthermore, Fig. 3 provides a visual comparison of the average slide representations along channels (i.e., features extracted using the pretrained ImageNet network) obtained using the BLOCK_3, GRID_4, and BOTH_4 attention modules, highlighting SIMM's ability to refine input features. In addition, Fig. 4 illustrates that the BLOCK attention module remains relatively stable across different window sizes, while the GRID and BOTH modules exhibit greater sensitivity. This may be due to the way spatial relationships are structured in them.

On the TCGA LUNG dataset, our best-performing models exhibit similar improvements, as shown in Tables 3 and 4.

These results underscore the effectiveness of our method in capturing spatial relationships among input patch features for accurate WSI classification. Notably, the BLOCK module consistently outperforms ABMIL with minimal or no additional computational cost, while also achieving comparable or superior performance to TRANSMIL despite being significantly more computationally efficient (94M vs. 614M FLOPs).

## 4 Conclusion

In summary, this paper presents the GABMIL model for weakly supervised WSIs classification in digital pathology to address a fundamental limitation of traditional MIL—ignoring the spatial information. To overcome this, we enhanced the ABMIL framework by incorporating spatial interactions between instances using BLOCK and GRID attention modules. Our approach effectively captures both local and global contextual information while maintaining computational efficiency of ABMIL. Through extensive evaluations on two publicly available datasets, our method consistently outperformed baseline MIL approaches, highlighting the importance of leveraging spatial correlations in weakly supervised WSI classification. Future work will explore the integration our SIMM within ABMIL to compute the attention weights while explicitly considering spatial relationships among patches.

## Acknowledgments and Disclosure of Funding

This work was done as a part of the IMI BigPicture project (IMI945358).

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
