# OpenReview forum: "A Spatially-Aware Multiple Instance Learning Framework for Digital Pathology"
_MICCAI.org/2025/Workshop/COMPAYL — COMPAYL 2025_

### Official Review · Reviewer_mrFh · 2025-07-10
**Review - A Spatially-Aware Multiple Instance Learning Framework for Digital Pathology**

**Rating:** 4
**Confidence:** 3

**Review:**

Summary:
This manuscript outlines GAMBIL, a Global Attention-Based Deep Multiple Instance Learning model for the classification of pathology slides. The traditional AMBIL method does not take into account spatial information when predicting slide labels, a feature which is of great importance in pathology. Whilst transformer based models exist to address this issue, they have very heavy computational overheads. This work aims to leverage AMBIL, with its MLP structure in predicting slide labels to overcome the computation burden invoked by transformer based models. The authors note improved model performance compared to AMBIL and TransMIL in two distinct cohorts, demonstrating the models utility and the importance of spatial information in pathology slide label predictions.

This work presents a novel framework for combining spatial context with existing prediction models. Despite these findings, the writing of the manuscript is flat and does not accurately convey the importance of the findings to the reader.

Strengths:
1. The GAMBIL model demonstrates consistent improvements in slide labelling compared to the AMBIL and presents improved or comparable performance when compared to TransMIL.
2. This model architecture includes spatial resolution, which is important for contextualising pathology.
3. The framework outlined in the manuscript significantly reduces computational overheads compared with TransMIL, and is comparable with AMBIL computational costs.

Weaknesses:
1. The introduction within this work outlines the problems faced, however not enough attention is brought to the high computational burden induced by transformer based predictions. As the technology exists to solve the spatial problem with MIL models, greater care is needed to highlight the impact and novelty of this work. It feels as though the authors undersell their work throughout the manuscript, leading the reader to question its importance, novelty and relevance. This should be addressed.
2. The authors do not note the size of their training and evaluation datasets for these models. The rationale for using 50 epochs is also not mentioned. Further expansion of the methods section is needed for clarity.
3. A more detailed figure legend for Figure 2 would improve clarity.
4. Author citations are included outside parentheses. Please check the referencing guidelines.

---

### Official Review · Reviewer_dcH7 · 2025-07-13
**Potentially interesting idea but lacking external validation**

**Rating:** 3
**Confidence:** 5

**Review:**

# Summary

The authors propose GABMIL, an extension of the classical Attention‑based Multiple Instance Learning (ABMIL) framework whole‑slide‑image (WSI) classification. GABMIL explicitly injects spatial context between patches through  BLOCK and GRID interaction modules (inspired by MaxViT). Compared with plain ABMIL, the method delivers up to 2 pp AUC and 7 pp AUPRC gains on TCGA‑BRCA and TCGA‑LUNG while maintaining computational efficiency; it also matches or outperforms the much heavier TransMIL baseline.

# Strengths

- The authors study an important problem in computational pathology: how to effectively incorporate spatial information into multiple instance learning. Most contemporary approaches tend to ignore the spatial information (although there are some MIL methods that do incorporate spatial information).
- GABMIL is computationally efficient, incurring only a small extra cost compared to classic ABMIL.
- The authors promise to release their code.

# Weaknesses

- A significant flaw in the study is its lack of external validation. External validation is the norm in the field of computational pathology. The tasks chosen by the authors (breast cancer classification and lung cancer classification) are common in the literature. Yet, in the literature, they are commonly evaluated using external validation datasets, e.g. training on TCGA-BRCA and testing on CPTAC-BRCA (all of these are open datasets). Without external validation, it is not possible to assess the significance of these results.
- The paper is missing key related work regarding MIL frameworks that employ spatial information, e.g. [4,5]. The authors should clarify the differences between their method and other MIL frameworks that also incorporate spatial information. Furthermore, some of the existing methods should be used as baselines.
- “These results underscore the effectiveness of our method in capturing spatial relationships among input patch feature for accurate WSI classification.” The fact that the authors were able to achieve marginally better results with their method does not mean that these gains in performance are due to the model being able to capture spatial relationships. Another explanation could be that the BLOCK and GRID attention modules simply add noise to the features, effectively acting as a regularisation mechanism.
- The results table is lacking a mean baseline (i.e. using ABMIL but weighting each patch equally).

# Detailed comments

- “However, current MIL methods often overlook interactions between distinct instances.” I disagree, as modern established MIL methods employ transformers which do model interactions between instances [7,8,9].
- “The flops are measured with 120 instances per bag.” This number seems very low. Usually, there are thousands of instances per bag (patches per whole slide) at 10x magnification (which is the magnification used in the paper). Please explain why there are only 120 instances per bag for computing flops. Also, please report the average number of instances per bag during training/inference.
- Please elaborate on the choice of self-supervised ResNet feature extractor (Kang et al). Various benchmarking studies on pathology feature extractors have shown that newer extractors such as H-optimus-0, Prov-GigaPath, and UNI consistently outperform Kang et al. [10,11,12]. Even Kang et al’s DINO model outperformed the ResNet model in Kang et al’ paper.
- “In summary, this paper […] address[es] a fundamental limitation of traditional MIL — ignoring the spatial information”. This is not a fundamental limitation of MIL, as there are a number of MIL frameworks that incorporate spatial information [6,7].
- Citations are improperly formatted (e.g. “Aeffner et al. (2019)” instead of “(Aeffner et al., 2019)”).
- Sections 2.2.1 and 2.2.2 begin with a lowercase and incomplete sentence

[3] https://arxiv.org/abs/2407.17689

[4] https://www.sciencedirect.com/science/article/pii/S1361841524001774

[5] https://arxiv.org/abs/2305.10552

[6] https://arxiv.org/abs/2305.10552

[7] https://arxiv.org/abs/2106.00908

[8] https://www.cell.com/cancer-cell/fulltext/S1535-6108(23)00278-7

[9] https://www.nature.com/articles/s41596-024-01047-2

[10] https://arxiv.org/abs/2408.15823

[11] https://arxiv.org/abs/2311.11772

[12] https://www.nature.com/articles/s41467-025-58796-1

---

### Official Review · Reviewer_oCmm · 2025-07-15
**ABMIL-based spatial context framework for WSI classification**

**Rating:** 3
**Confidence:** 4

**Review:**

Summary:
- The authors propose a framework for WSI classification, based on ABMIL, integrating spatial context between patch embeddings, while preserving computational efficiency.

Strengths:
- computationally efficient model, while keeping performance;
- well-written manuscript.

Weaknesses:
- Figure 2 is not fully understandable. What is represented in the images? What does the color grading mean? How does it highlight the ability of SIMM to refine the input features? From the examples provided, it’s not clear that the model is learning any relevant spatial relations of the embeddings;
- The authors present the AUPRC scores using the ImageNet pretrained feature extractor, while from the tables, the best results are obtained with the self-supervised feature extractor. Additionally, the curves should include the standard deviation values;
- The authors do not mention how the data was split;
- Experiments lack external validation.